# Environmental and Agronomical Factors Limiting Differences in Potato Yielding between Organic and Conventional Production System

Krystyna Zarzyńska * [ID], Cezary Trawczyński [ID] and Milena Pietraszko

Plant Breeding and Acclimatization Institute, National Research Institute in Radzików, Jadwisin Division, Potato Agronomy Department, 05-140 Serock, Poland
* Correspondence: k.zarzynska@ihar.edu.pl

**Abstract:** This paper presents the results of the authors' own research and literature research on the impact of selected environmental and agronomical factors on the yield of potato grown under the organic system and the possibility of increasing the yield. The results are based on research conducted for several years at the Institute of Plant Breeding and Acclimatization in Jadwisin, Poland. The influence of factors such as soil quality and climatic conditions, selection of varieties, seed potato preparation, irrigation of plantations, complementary fertilization, and protection against the late blight was described. The aim of this work was to indicate which of these factors affect the yield increase and to what extent. It was stated that it is possible to increase the yield of potato tubers grown under the organic system through all of the proposed treatments. In our studies, using drip irrigation and complementary fertilization had the greatest effect (25.5% and 19%, respectively). Seed potato presprouting had a smaller influence (4.3%) on the final tuber yield. In the years with high pressure of the pathogen *Phytophthora infestans*, the selection of cultivars with high resistance was very important. Most of the agronomical treatments not only improved the total yield of tubers, but also increased the share of tubers with a larger diameter. A very high variability of potato yielding depending on weather conditions and a selection of cultivars was emphasized. We can say that a proper agronomical practice carried out on an organic potato plantation can largely eliminate the yielding gap between a conventional and an organic system.

**Keywords:** potato; organic; conventional system; yield; agronomical treatments

## 1. Introduction

In the literature, there are many considerations regarding the differences in the yield of plants cultivated in the organic and conventional systems. The question is often asked whether organic farming can guarantee food for people [1,2]. Many researchers have tried to answer this question by evaluating the differentiation of yields in organic and conventional agriculture. Recent data for organic yield reduction are estimates between 9 and 25% [3]. Numerous studies and practices show that the organic production system gives lower, more variable yields than systems using synthetic fertilizers and chemical plant protection products [4]. However, the difference in yield between these systems is dependent on the crop species, with root crops showing a greater difference than cereals.

The potato (*Solanum tuberosum* L.) is one of the staple foods of modern western civilization and it is becoming increasingly important in developing countries. The potato is the fourth most important food crop in the world ranking at about 400 million tons per year [5]. Potato is not an easy crop to grow in organic production systems due to the high threat from diseases and pests. Out of the factors which limit yield of potato crops grown in organic production systems, soil nutrient deficits and the restriction on use of pesticides are two of the most common. The main pests worldwide are late blight caused by (*Phytophthora infestans*), and Colorado potato beetle (*Leptinotarsa decemlineata*). At the

present stage, the problem of fighting the beetle is less troublesome due to the availability of effective biological agents for protection against this pest.

Each species has its limiting factor. The rule Sprengel's and Liebig's stating that yield losses are not determined by the sum of the factors affecting the yield, but by the factor having the greatest impact applies here. In the case of cereals, which are characterized by a rapid growth rate in the initial period, and potato, which has a high demand for nutrients in the short term, the availability of these nutrients is the main limiting factor. In the case of legumes, these are mainly weeds and pests.

The relatively large difference in yield between organic and conventional potato production is mainly attributed to inadequate protection against late blight caused by *P. infestans* [6]. Weaker development of the above-ground part of plants cultivated in the organic system has a direct impact on the yield of tubers and its structure.

As previously mentioned, the yield gap for tubers is often greater than for cereals, but it is are also more variable [7]. Of all crops, root crops have the second largest yield gap between the organic and conventional systems [8]. Comparing all European countries as regards 21 aspects, de Ponti et al. [9] found that potato yields from organic plantations accounted for only 70% of conventional crops. If potato was classified as a vegetable, where the difference is 33%, in the case of potato, the difference was almost 30%, as presented by Ponisio et al. [10]. As already emphasized in the case of potato, the main factor limiting the yield in organic production is the availability of nutrients, followed by pathogens. Möller et al. [11] found that 48% of the gap in organic potato yields can be attributed to reduced fertilization, especially nitrogen, and 25% to diseases, mainly late blight.

The aim of the work is to indicate the environmental and agrotechnical factors affecting the yield of potato tubers grown in the organic system and to show which of these factors can reduce the differences in yield between the organic and conventional systems.

## 2. Materials and Methods

The results presented in the paper are based on research conducted for several years (2005–2020) at the Institute of Plant Breeding and Acclimatization in Jadwisin on the cultivation of potato plants in various production systems, including the organic system. In the long-term studies, factors such as environmental conditions, including soil quality and weather conditions of the growing season, variety diversity and agronomical factors such as method of seed potato preparation, fertilization, irrigation, and differentiated protection against diseases were taken into account. Some of the research was carried out in two locations: Jadwisin—central Poland 52.51° N and 21.07° E and Osiny-south-eastern Poland 51.42° N and 21.97° E (Figure 1) on two different soil complexes: in Jadwisin—good rye, in Osiny—very good rye. Crop rotation adapted to the soil conditions was used in each locality. The research was conducted in accordance with the requirements applicable to organic farming. Detailed methodologies for individual experiments are described when discussing these studies.

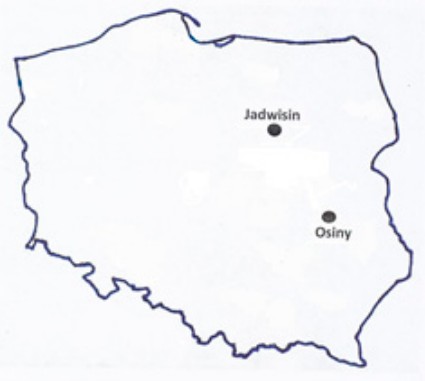

**Figure 1.** Places of investigation (Jadwisin, Osiny) in Poland.

Statistical analyses of the results were performed with an analysis of variance using Statistica software (StatSoft, Krakow, Poland). The significance of the sources of variation was tested with a Fisher–Snedecor test, and the significance of differences was assessed using Tukey's test.

In addition to the results of own research, the results from the literature were also taken into account.

## 3. Results and Discussion

### 3.1. Influence of Climatic and Soil Conditions on the Yield of Potato Tubers Cultivated in an Organic System

The impact of climatic and soil conditions on the yield of tubers was presented on the basis of research conducted in the years 2005–2016 in two sites located in different regions of Poland, with different soil and climatic conditions. One experimental site was the Experimental Station Osiny and the other was Jadwisin. The research was conducted in 3-year cycles. While evaluating the yield of tubers, reference was made to the conditions of the vegetation period prevailing in particular localities. They are described by the Sielianinov coefficient, which takes into account temperature and precipitation (Table 1).

**Table 1.** The vegetation period conditions expressed by the Sielianinov hydrothermal coefficient in the years of research for Jadwisin and Osiny, 2005–2016.

| Year | Site | Sielianinov Hydrothermal Coefficient (K) | Year | Sielianinov Hydrothermal Coefficient (K) |
|---|---|---|---|---|
| 2005 | Jadwisin | 1.0 | 2011 | 1.8 |
| | Osiny | 1.3 | | 1.5 |
| 2006 | Jadwisin | 1.1 | 2012 | 1.5 |
| | Osiny | 1.3 | | 1.2 |
| 2007 | Jadwisin | 1.7 | 2013 | 1,8 |
| | Osiny | 1,7 | | 0.8 |
| 2008 | Jadwisin | 1.1 | 2014 | 0.9 |
| | Osiny | 1.4 | | 1.8 |
| 2009 | Jadwisin | 1.2 | 2015 | 0,6 |
| | Osiny | 1.2 | | 0.9 |
| 2010 | Jadwisin | 2.0 | 2016 | 1.3 |
| | Osiny | 1.5 | | 1.2 |

K = 0–0.5 drought, K = 0.6–1.0 dryness, K > 1 wet conditions.

A very large diversification of tuber yield was found in both sites in individual years. In the first years of the experiment in Jadwisin, on lighter soil, yields were much lower than in Osiny. Particularly large differences concerned the years 2005 and 2006. Differences in yield in favor of heavier soil were found until 2006. After this period, these differences were smaller, and yields in Jadwisin were even higher than in Osiny. The greatest differences in favor of Jadwisin were recorded in 2008, 2009 and 2014. Higher yields in Osiny were obtained in 2013 and 2015. The yield was closely related to the weather conditions in a given locality. In 2013, very unfavorable conditions in Jadwisin contributed to a strong reduction in yields. However, 2015 was more favorable for obtaining higher yields in Osiny. The average yield from both organic plantations was similar and amounted to 23.7 t·ha$^{-1}$ in Jadwisin and 21.7 t·ha$^{-1}$ in Osiny. On average, in both localities, the highest yields were obtained in 2016, 2012 and 2009, and the lowest in 2005, 2006 and 2013. Greater yield stability was noted on the heavier soil (Figure 2).

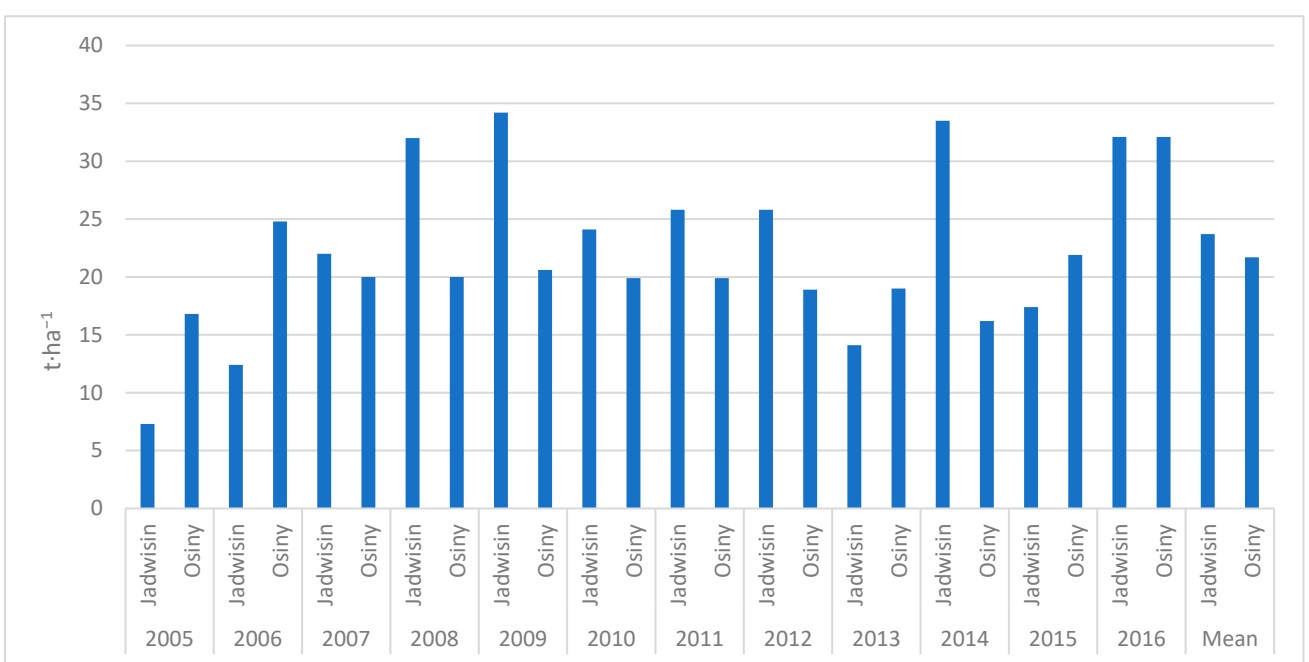

**Figure 2.** Tuber yield depending on site and years of growing (mean for 30 varieties).

The assessment of the impact of environmental factors on the yield showed that the years of research have the greatest impact. This is related to the development of diseases, especially potato blight, which causes the greatest losses in the crop.

The analysis of the weather course in particular years confirms the principle that "blight" years, i.e., with a lot of rainfall, are years of high potato yields. This mainly applies to the conventional system, where the use of mineral fertilization and pesticides is allowed. In the organic system, the situation is more difficult, which makes the yield in this system more dependent on climatic conditions, which is confirmed by the research of Zarzyńska and Pietraszko [12]. Although copper fungicides are allowed to protect potato plants against late blight in organic crops, their effectiveness is not as high as that of other fungicides. A serious factor limiting the yield is the lack of rainfall or its uneven distribution. The main reason for the drastically low yields in 2005 in Jadwisin, on lighter soil, were long periods of deep drought. Additionally, in 2006, the lack of precipitation in June and July adversely affected yields in both localities. A similar situation occurred in 2013 and 2015, when a very wet spring (especially in Jadwisin) in full vegetation, i.e., in July, was followed by a drought. The most favorable weather conditions in both sites were in 2016 and this year the highest yields were obtained.

The impact of extremely different weather conditions on the yield of potatoes grown in the organic and conventional system in Jadwisin was demonstrated in 2012 and 2013. In 2012, both the amount and distribution of precipitation, as well as the air temperature, were favorable for the growth and development of potato plants; however, in 2013, these were very unfavorable. After spring excess rainfall and low air temperatures, a large shortage of water was noted later. A significantly higher yield was obtained in the conventional system and compared to the years of research in 2012. It should be noted that the differences in the years were higher than between the production systems, and the plants grown in the organic system reacted with a greater decrease to unfavorable weather conditions. The decrease in yield between years was 65%, and between the production systems 50% [12].

### 3.2. Influence of Cultivar on the Yield of Potato Tubers Cultivated in an Organic System

Within 12 years, 30 potato varieties belonging to different groups of maturity were tested. Their differentiation in terms of yield was significant. The highest yields (average for the years and the place of cultivation) were obtained for: Jurata, Tajfun, Ursus, Ignacy,

Michalina, Vitara, Owacja, Agnes, Finezja, Oberon and Malaga (over 25 t·ha$^{-1}$), and the lowest for Drop and Gracja (below 15 t·ha$^{-1}$). The significance of the interaction of cultivar and place of cultivation was not proven, but a varied response of cultivars was observed. In Jadwisin, the largest number of cultivars yielded over 24.0 t·ha$^{-1}$ in Osiny in the range of 21–24.5 t·ha$^{-1}$. Some cultivars achieving the highest yields in Osiny gave one of the lower yields in Jadwisin and vice versa. Cultivars with extremely different yield levels in both localities were: Bartek, Berber and Vitara (Figure 3).

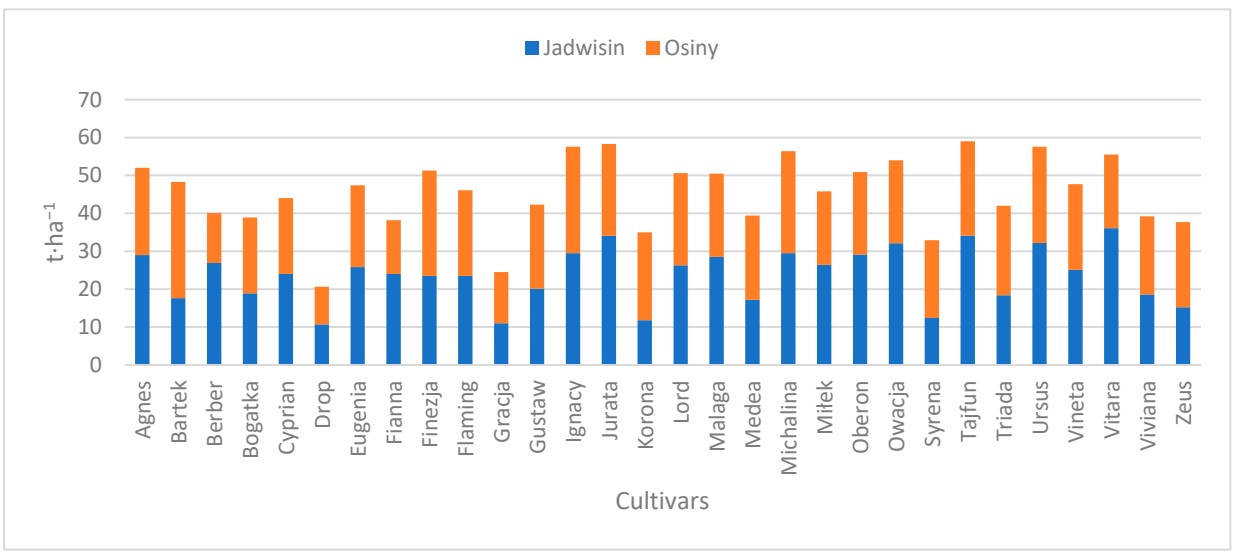

**Figure 3.** Yield of cultivars depending on site of growing (mean for 3 years).

The yielding of several cultivars from the same groups of maturity originating from Polish and foreign breeding was also compared. Significant differences in tuber yield and its structure were found depending on their origin. Polish cultivars yielded higher than foreign. They were also characterized by a better yield structure, i.e., they gave a higher marketable yield and a higher yield of large tubers (Table 2).

**Table 2.** Tuber yield and its structure of Polish and foreign cultivars cultivated in an organic system (Jadwisin, 2008–2010).

| Maturity Group | Cultivar | Country of Origin | Total Yield (t·ha$^{-1}$) | Yield of Big Tubers (>60 mm) (t·ha$^{-1}$) |
|---|---|---|---|---|
| Very early | Miłek | Poland | 22.9 | 1.9 |
| | Berber | The Netherlands | 20.1 | 2.0 |
| Early | Owacja | Poland | 27.0 | 5.3 |
| | Vitara | Germany | 27.8 | 5.5 |
| Mid early | Tajfun | Poland | 29.5 | 5.0 |
| | Agnes | Germany | 26.1 | 4.4 |
| Mid late and late | Ursus | Poland | 28.8 | 4.6 |
| | Fianna | The Netherlands | 19.1 | 1.0 |
| Mean for Polish cultivars | | | 27.1 | 5.7 |
| Mean for foreign cultivars | | | 23.3 | 3.2 |
| LSD | | | 3.1 | 1.9 |

LSD—least significant difference.

The presented results show that the appropriate selection of varieties and adapting them to the soil and climatic conditions in a given region plays a very important role.

Very early and early varieties are generally characterized by low resistance to the fungus-like organism *P. infestans* causing late blight; however, in organic cultivation, they are often found due to the faster rate of yield accumulation and the possibility of "escape" from the blight. Late mature cultivars are more susceptible to pathogen infection later in development, but they also have greater disease resistance [13–17]. It should be also emphasized that regional, domestic varieties turn out to be better than foreign ones in organic production [18,19].

Cultivation in organic conditions requires adaptability and greater stability of varieties [20]. The most important features of varieties suitable for organic farming are: rapid development in the initial growth period, large root system, high resistance to major diseases, low fertilization requirements [16,20,21]. These features occur mainly in older, native varieties [22]. On the other hand, numerous experiments suggest that modern varieties are well-adapted to harsh environmental conditions and show a higher level of disease resistance. The new varieties are also more acceptable for consumers [23].

*3.3. Influence of Selected Agronomical Treatments on the Yield of Potatoes Cultivated in Organic System*

As has been repeatedly emphasized, the yield from the organic system is lower than from the conventional which is why new solutions are constantly being sought to improve the efficiency of cultivation in the organic system and increase yields. These treatments include the proper preparation of seed potatoes, irrigation of the plantation, the use of additional fertilization, and the improvement of protection methods against pests. In our research, we assessed the effect of these treatments on the tuber yield.

The Effect of Seed Potato Preparation on the Yield of Potato Tubers Cultivated in an Organic System

The research was carried out in 2008–2010 in Jadwisin. The effect of seed potato presprouting on tuber yield and its structure was assessed. This treatment was used for 4 weeks. The seed potatoes were sprouted at 15 °C in the light. Eight potato cultivars belonging to different groups of maturity were tested: Berber, Miłek (very early), Owacja, Vitara (early), Tajfun, Agnes (medium early), Fianna, Ursus (medium late and late). Presprouting of seed potatoes increased the total yield of tubers, but the differences were not statistically significant. The yield increase was 4.3%. The treatment significantly increased the yield of large tubers, i.e., over 60 mm of 29.7% (Table 3).

**Table 3.** The influence of seed potatoes presprouting on tuber yield and tuber size distribution (Jadwisin, 2008–2010).

| Seed Potatoes Preparation | Total Yield (t·ha$^{-1}$) | Yield of Small Tubers (<35 mm) t·ha$^{-1}$ | Yield of Medium Tubers (35–60 mm), t·ha$^{-1}$ | Yield of Large Tubers (>60 mm) t·ha$^{-1}$ |
|---|---|---|---|---|
| Presprouted | 32.2 a | 1.9 a | 26.6 a | 3.7 a |
| Without presprouting (control) | 30.8 a | 1.8 a | 25.5 a | 2.6 b |
| Difference in relation to control (%) | 4.3 | −5.3 | 4.1 | 29.7 |

a, b—mean values indicated by the same letters are not statistically significant at the 0.05 level by Tukey's test.

Presprouting of seed potatoes plays a particularly important role in organic production, because it causes a faster development of plants, which promotes the so-called "escape from the blight", and shifts the vegetation to a period of better sunlight, which increases the efficiency of photosynthesis. Presprouting also causes better development of the root system, which facilitates the uptake of water and nutrients and improves the resistance of plants to viruses. It also accelerates the maturation of tubers, which is conducive to their better storage. The positive effect of priming on plant development and tuber yield in organic potato cultivation was previously described in the work of [24–27]. Presprouting

of seed potatoes has the greatest impact on accelerating the accumulation of the crop and its greatest effect is visible in the early dates of harvest. In the final harvest, this effect often disappears. This is confirmed by research of Karalus and Rauber [27] carried out on several potato cultivars grown in an organic system. In their experiment, in the trial dates of harvest, i.e., in July, presprouting significantly increased the tuber yield in relation to the control. There was no such effect at the final harvest. Presprouting reduced the percentage of undersized tubers and significantly increased the percentage of oversized tubers. A significant increase in yield due to presprouting was seen when the plants were damaged by Colorado beetle or late blight in the early growing season. Therefore, preprouting can be an important procedure to increase yield, mainly in systems without the use of pesticidies. For example, organic growers in temperate regions avoid severe damage from late blight by planting early maturing presprouted potato varieties early in the growing season so that tubers have grown to a reasonable size by the time late blight becomes pervasive.

The effect of accelerating of sprouting and plant emergence on their faster development and reducing diseases depends to a large extent on the nitrogen content of the soil. In the studies of Möller and Reents [28] in conditions of high N abundance, the acceleration of plant development and tuberization by presprouting resulted in an increase in yield by 18–23%, and the use of varieties with a short vegetation period increased the yield (by reducing the occurrence of late blight) by 0–21%. Under conditions of low N abundance, these strategies have less impact on increasing the yield. This is due to the fact that with low N abundance, plant growth is very weak and usually ends before the onset of the greatest severity of the disease. In Hagman's research [26], the final yield of presprouted tubers was 7–24% higher compared to control. However, this effect decreased over time during the growing season.

### 3.4. Influence of the Complementary Fertilization on the Yield of Potato Cultivated in an Organic System

The priority issue in the potato organic production system is to maintain proper soil fertility, because the potato is a crop with high nutritional requirements. To produce 1 ton of tubers, it needs 4–5 kg of N, 0.7–0.8 kg of P, 5–6 kg of K and significant amounts of microelements. The basic source of these nutrients and humus, which determines the high fertility of the soil, is the introduction of an appropriate amount of natural or organic fertilizers produced on the farm into the soil. The use of farm-produced compost or manure obtained from animals, as well as plowing in straw and biomass of catch crops, as well as appropriate control of the decomposition of organic matter, allow you to provide plants with the necessary nutrients. The effectiveness of organic fertilization used in the organic system depends on many factors, both related to the type of fertilizer used (dose, date) as well as soil conditions or the course of weather during the vegetation period of plants. Mainly unfavorable combination of soil and climatic conditions may contribute to certain difficulties in the decomposition of the applied fertilizers and the uptake of nutrients from it. This may cause various types of stress and disorders in the development of potato plants, which should encourage the use of specific fertilization treatments to counteract these unfavorable changes [29–33].

### 3.4.1. The Use of Microbiological Preparations in the Organic Potato Cultivation

The application of microbiological preparations to the soil increases its biological activity and eliminates putrefactive processes, dissolves mineral compounds that are difficult to access for plants, improves fertility and structure, which may have a beneficial effect, among others, on potato yield. In the years 2004–2006 in our institute, field tests were carried out on light soil, the purpose of which was to determine the effect of the UG max Soil Fertilizer on the yield of potato tubers. The UG max Soil Fertilizer was applied to the straw immediately before its plowing in a dose of 3 L·ha$^{-1}$ dissolved in 300 L·ha$^{-1}$ of water. After the use of straw in combination with the UG max Soil Fertilizer, by 9.2% higher yield of potato tubers was obtained compared to the use of straw alone (Figure 4 [34]).

The beneficial effect of UG max Soil Fertilizer was confirmed by Zarzecka and Gugała [35]. Soil Fertilizer UG max applied in different doses and dates contributed to an increase in the total yield and the yield of large potato tubers by 27.2 and 35.3%, respectively, compared to the control treatment. On the other hand, in the studies of Jabłoński [36] the UG max Soil Fertilizer in the dose of 1 L·ha$^{-1}$ increased the total yield by 12.2%, marketable yield by 15.1%, and large tubers by 20.3%. In the research of Kowalska [37], the effect of two microbial preparations (EM Farma Plus and UG max) applied to the soil, foliar or together was evaluated. The application of combined microbiological treatments (soil spraying before planting and 4 foliar treatments during the growing season) significantly increased the yield of potatoes and the share of marketable tubers in the yield. On the other hand, the application of commonly used natural and organic fertilizers such as cattle manure, green manure and compost of effective microorganisms (EM), in addition to increasing potato yields by 10 to 16%, reduced the infestation with fungal diseases [38]. In turn, Kołodziejczyk [39] showed that the use of microbiological preparations, BactoFil B10, Effective Microorganisms EM and UG max, led to an increase in the nitrogen content of the soil after plant harvest and lower uptake of this component, reducing the NUE, NUpE, NAE and NRF indicators. Another bacterial strain (*Bacillus cereaus*), in addition to increasing the availability of potassium in the soil by 42% and potassium uptake by tubers by 62%, significantly increased the total potato yield by 21% compared to control plants [40]. In many studies conducted to date, the use of a microbiological preparation has also resulted in an improvement in plant health during the growing season and tuber quality characteristics after the final harvest [41–43].

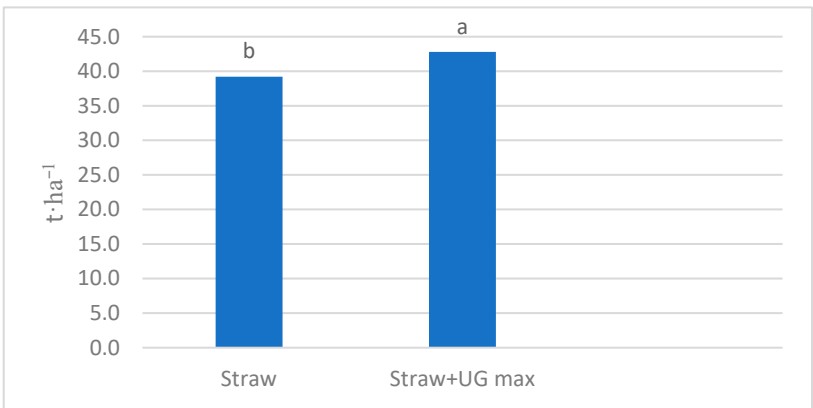

**Figure 4.** Effect of UG max Soil Fertilizer on potato tuber yield. a,b-mean values indicated by the same letters are not statistically significant at the 0.05 level by Tukey's test.

### 3.4.2. The Use of Humic Preparations as an Alternative and Supplement to Traditionally Used Forms of Natural and Organic Fertilizers in Organic Potato Cultivation

In farms without livestock, especially, but also in the conditions of animal production, there may be problems with insufficient amount or obtaining the appropriate organic biomass that maintains soil fertility at the appropriate level, which allows one to obtain a high and appropriate quality yield of potato tubers. An alternative in this regard may be the use of humic preparations containing humic acids, the main component of humus [44]. One of them is organic preparation Rosahumus. Humic acids in Rosahumus are obtained from leonardites, a mineral easily soluble in water with a content of 85% of this compound. In addition to humic acids, the preparation is a rich source of potassium ($K_2O$)—12% and iron (Fe)—0.6% (https://agrosimex.pl, accessed on 5 March 2023). In the authors' own research conducted in the years 2018–2020, humic preparation was applied to the soil before planting tubers, immediately before ridding, in the form of spraying with a water solution at a dose of 4.5 kg·ha$^{-1}$ dissolved in 300 l·ha$^{-1}$ of water. Under the influence of this preparation, a significant increase in tuber yield was obtained, which amounted to 2.5 t·ha$^{-1}$, which was 11.9% compared to the control, without the use of humic preparation

(Figure 5). In the study of Fatma et al. [45] humic acid added to irrigation water generally improved plant growth parameters and potato tuber yield. In addition, with the increase in the level of humic acids in the irrigation water, the vigor of plants increased and the quality of the crop improved. In other studies, it was proven that treatments with humic acid and algae extracts caused a significant increase in all morphological features of potato plants and its yield. After application of humic acid to the plants and spraying with a mixture of Alga 600 and Sea Force 2, significantly higher values of vegetative features were obtained compared to the control object. A significantly positive effect of humic acid and seaweed extracts and their interaction on all yield characteristics was obtained. After application of humic acid with Alga 600 and Sea Force 2 potato plants were characterized by the following values: number of tubers per plant 9.42, average tuber weight 82.49 g, crop yield 0.780 kg/plant and total yield per hectare 34.52 tones and dry weight of tubers at the level of 14.67%, while on the control object the values of these features were, respectively: 7.25; 73.10 g; 0.540 kg/plant, 23.88 tones and 13.37% [46]. Mon-hong et al. [47] found that humic acid increased the yield of tubers with water retention: 45%, 60% and 75% in the years 2014–2015, respectively, by 34.47–63.48%, 35.95–37.28% and 23.37–27.15%. In the research conducted by Suh et al. [48] after spraying potato plants with fulvic acid 50, 60 and 70 days after tuber planting, there was no significant difference in the number of tubers and total yield compared to the control. On the other hand, after applying humic acid to the soil in the dose of 40 and 80 $g \cdot m^2$ before planting potato tubers, an increase in the weight of large tubers was found. However, in the studies of Ekin et al. [49] after treating potato tubers with humic acid in combination with bacterial strain cultures (*Bacillus magatorium* and *Bacillus subtiles*), a significant increase in potato plants, tuber yield and improvement in its quality was obtained.

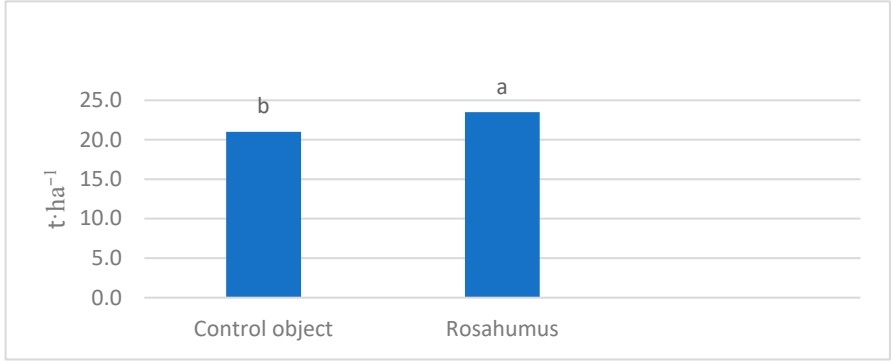

**Figure 5.** Effect of organic preparation Rosahumus on potato tuber yield. a,b-mean values indicated by the same letters are not statistically significant at the 0.05 level by Tukey's test.

### 3.4.3. Complementary Foliar Feeding of Potato Plants with the Use of Nutritional Biological Agents

It has happened quite often in recent years that during the vegetation of plants, the combination of weather factors, such as heavy rainfall alternating with periodic soil drought, high air temperature, etc., causes certain difficulties in the uptake of nutrients from the soil. This can cause various types of stress and disturbances in plant development [33]. Due to the relatively long period of harvesting and the large mass of the crop produced by the potato, efforts should be made to ensure its optimal supply with easily digestible nutrients throughout the entire vegetation period. Therefore, foliar application should be a form of supplementing ingredients during plant growth, especially in critical periods with fast-acting biological agents that provide the necessary ingredients and bioactive substances with a biostimulating effect. A very broad formula when it comes to foliar application are biostimulating preparations: humic acid, fulvic acid, protein hydrolysates, seaweed extracts, chitosan, inorganic compounds or beneficial microorganisms [50]. Based on the earlier research, it can be concluded that the proposed division does not exhaust the

types of all biostimulating preparations used, which also include, for example, biological agents in the form of nanoparticles [51–53]. Biological agents of this type are produced using pro-ecological and innovative production technologies, and some of them were the subject of own research conducted on potatoes. In the studies carried out in 2018–2019, the effect of foliar feeding of potato plants with amino acid biostimulators, Naturamin Plus and Naturamin WSP used in the BBCH 19 and 39 phases on potato yielding, was determined. Under the influence of Naturamin Plus biological agent, tuber yield increased by 15.9%, and Naturamin WSP biological agent by 19.0% compared to the control, without foliar feeding, and this difference was not statistically proven (Figure 6) A greater increase in tuber yield under the influence of these foliar biological agents was obtained in 2018, which was characterized by a higher air temperature during the growing season than in 2019 [54]. A positive effect on the yield of potato tubers, but very diverse depending on the type of biostimulating biological agents with amino acids used, ranging from 3 to 36%, was obtained in other studies [55–58]. As in our research, a better yielding effect was generally obtained in years with unfavorable weather conditions during the plant vegetation period, which was caused by periodic drought or excess precipitation and high air temperature [46]. In 2018–2020, the effect of foliar feeding of potato plants with multi-component biological agents in the form of nanoparticles: Herbagreen Basic and Nano Active Forte on potato yielding was determined. Biological agents were applied twice during the growing season of potato plants, in a dose of 2 kg·ha$^{-1}$, in the BBCH 20 and BBCH 59 phases. Under the influence of these compound biological agents, a similar increase in potato tuber yield of 3.0 t·ha$^{-1}$ was obtained, which amounted to 11.5% in compared to the control, without foliar feeding (Figure 7) [59]. The beneficial effect of foliar application of nanopaticles biological agents has been demonstrated in other studies, and the increase in tuber yield ranged from 13 to 32% [60–63].

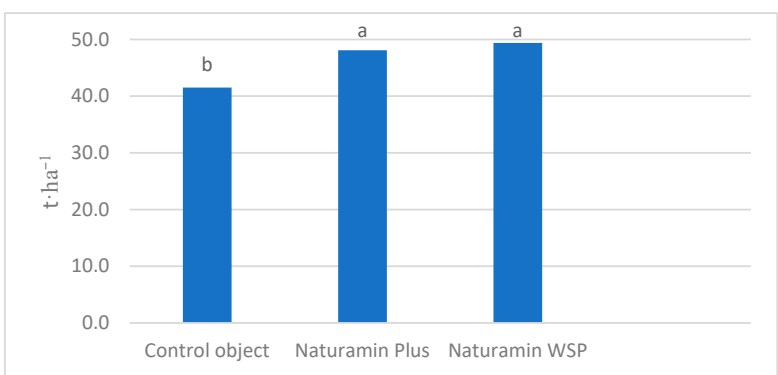

**Figure 6.** The effect of foliar biological agents based on amino acids on potato tuber yield. a,b-mean values indicated by the same letters are not statistically significant at the 0.05 level by Tukey's test.

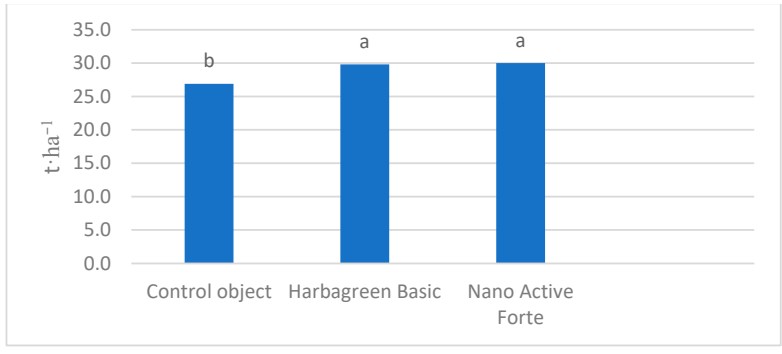

**Figure 7.** The effect of multi-component nanoparticle biological agents on potato tuber yield. a,b-mean values indicated by the same letters are not statistically significant at the 0.05 level by Tukey's test.

Table [4] presents a synthetic summary the effect of individual preparations on the yield increase in relation to the control.

**Table 4.** Yield increase under the influence of the tested natural preparations (Jadwisin 2004–2020).

| Bilogical Agent | Increase of Total Tuber Yield [%] |
|---|---|
| UG max | 9.2 |
| Rosahumus | 11.9 |
| Naturamin Plus | 15.9 |
| Naturamin WSP | 19.0 |
| Herbagreen Basic | 10.8 |
| Nano Active Forte | 11.5 |

*3.5. Influence of Irrigation of a Plantation on the Yield of Potato Tubers Cultivated in an Organic System*

Research on the irrigation of an organic potato plantation has been carried out in the years 2014–2015 in Jadwisin. Drip irrigation was used in both years. In 2014, the treatment was used twice in doses of 19.8 and 7.8 mm. In 2015, the rainfall deficit was greater; therefore, the plantation was irrigated four times in doses: 15.1, 18.5, 20 and 16.3 mm. The total amount of water supplied in 2014 was 37.6 mm and in 2015 it was 69.9 mm. Eight potato cultivars from different groups of earliness were tested in the experiment.

Irrigation of the plantation significantly increased the total yield and improved its tuber size distribution. The yield increase was 25.5%. The treatment significantly reduced the share of the smallest tubers and increased the share of large tubers (Table [5]).

**Table 5.** Influence of irrigation on tuber yield and tuber size distribution.

| Combination | Total Yield (t·ha$^{-1}$) | Share of Small Tubers (<35 mm) (%) | Share of Medium Tubers (35–60 mm) (%) | Share of Big Tubers (>60 mm) (%) |
|---|---|---|---|---|
| With irrigation | 33.3 a | 3.7 b | 82.1 a | 14.2 a |
| Without irrigation (control) | 24.8 b | 5.4 a | 87.5 a | 6.1 b |
| Difference in relation to control (%) | 25.5 | −30.9 | −6.2 | 56.7 |

a,b-mean values indicated by the same letters are not statistically significant at the 0.05 level by Tukey's test.

The influence of irrigation on tuber yield is well-known. Its importance is confirmed by many authors [25,64–67]. They emphasize that the lack of water not only reduces the yield but also causes its diminution [68–73]. This is especially important in organic production. Drought is a severe environmental stress limiting agricultural production in many countries. However, water availability for agriculture production is being reduced as a consequence of global climate change, and growing demand for other uses. Therefore, great emphasis is placed on water management based on plant physiology, with the aim of increasing water use efficiency. Potato is a species with high water requirements, and increasingly frequent shortages of water result in the final yield not being satisfactory. Deficiency of rainfall or its uneven distribution is the cause of inhibition of plant development, which affects the yield and its structure, and mainly the lower share of marketable tubers. The potato plant has different water needs during the growing season [74–76]. It needs the most water during tuber formation. In cultivars with a short vegetation period, this time is from June to early July. Late mature cultivars have the greatest demand for water from the second decade of June to the end of August [76,77]. Irrigation is the solution to water scarcity. Unfortunately, in smaller farms growing potatoes in an organic system, it is rarely used. The shortage of water combined with the inability to use mineral fertilizers can lead to large decreases in yield. Our experience shows that in practice, organic fertilization is often used on organic farms, mainly manure without irrigation. The use of nitrogen, especially in years with a large water shortage, is then very low [67]. In the study of Fatma et al. [78], four levels

of irrigation were used (100, 75, 50 and 25% ETO, and four types of organic fertilization (control, cow, sheep, and chicken). Morphological and physiological parameters of plants and tuber yield were studied. The best results were obtained using chicken manure and irrigation at 100% ETO. The conclusion of this research is that if the farmer's goal is to maximize yield, irrigation should be applied at 100% reference evapotranspiration.

However, the use of irrigation in organic potato cultivation can have negative consequences. Sprinkler irrigation of plantations increases the risk of late blight and may leach nutrients from the rhizosphere to the deeper layers of the soil. The drip irrigation used in our research seems to be the ideal solution.

*3.6. Influence of Treatments Limiting the Development of Late Blight on the Yield of Potato Grown in an Organic System*

As mentioned before, one of the most dangerous potato pathogens is the fungus *P. infestans.* causing potato blight. In organic production, the fight against this pathogen is particularly difficult. Our long-term observations show that on an organic plantation the appearance of the first symptoms of the blight is later and its rate of spread is slower than on a conventional plantation. This is related to the weaker development of plants, which is a consequence of restrictions on the use of mineral fertilizers and pesticides. Particular importance is the availability of nitrogen, both in the first weeks after emergence and during the flowering period when tuber formation begins [79–83].

In the experiment conducted in 2008–2010 by Zarzyńska and Szutkowska [7], four cultivars of potato were grown in the organic and conventional systems. The morphological and physiological parameters of plants, such as height, leaf mass, stem mass, LAI, PAR and SPAD, were evaluated. The rate of spread of late blight as well as the yield and its structure were also determined. Significant differentiation of most parameters has been proven. The values of plant productivity indices were lower in plants growing in the organic system. The result of these differences was a 22% lower tuber yield and a smaller tuber size. The positive side of the weaker development of plants in the organic system was the later appearance of blight symptoms and a slower rate of its development. As we know, this rate depends on the microclimate in the canopy [84]. In the case of a less developed above-ground part of plants, the canopy is more aerated and the development of the disease is slower [85].

These results show that in order to cause a less favorable climate for the late blight development, the density of plants in the canopy can be reduced. In our research conducted in 2012, we used different planting densities, shaping the canopy architecture accordingly, but the effects of this treatment were not visible due to the lack of disease occurrence (authors' observation).

Studies conducted in the UK and the Netherlands tested strategies such as sprouting and early planting of seed potatoes, which were supposed to accelerate the development of plants and cause escape from the blight, and diversified plant configuration, so as to create a less favorable microclimate for the development of the disease. Both sprouting and early planting shifted harvesting to less severe disease symptoms, especially in years when blight onset was early. The varied canopy architecture had no influence on the development of the late blight [86].

In the years 2010 and 2011 with different pressures of *P. infestans* (2010—low and 2011—high preasure) the research was carried out on the development of late blight in two production systems, i.e., organic and conventional. Copper fungicides were used in the organic system, and chemical fungicides in the conventional one. The protection in the conventional system was carried out on two levels, i.e., limited (3 treatments) and intensive (7 treatments). The study was carried out on 10 cultivars with different resistance to *P. infestans* in three earliness groups: very early and early, medium early and late. In the year with low pathogen pressure (2010), the rate of disease spread in both systems was similar. In the case of varieties with higher resistance to the pathogen, the spread of the disease was even slower in the organic system than in conventional. However, when

comparing the organic and conventional systems, but with intensive protection, a slower rate of disease development was noted in the conventional system. The situation was slightly different in 2011 with favorable conditions for the development of the late blight. Under these conditions, the rate of disease spread was significantly faster in the organic system compared to the conventional. Differences in the rate of spreading the disease in the conventional system at different levels of protection intensity were much smaller, although statistically significant. The maturity group of cultivars and their resistance to the pathogen were also important. However, this concerned only the group of cultivars least resistant to pathogen, i.e., very early and early, and the group of other cultivars. In the group of very early and early cultivars, the spread rate of the disease was significantly higher. Differences between the group of medium–early varieties and the group of late maturity varieties were not significant [17].

The yield of tubers depended significantly on all the studied factors, i.e., the production system, years of research and earliness groups of potato cultivars. The average yield of tubers from two years of research in the organic system was 23.6 t·ha$^{-1}$ in conventional with limited protection 44.8 t·ha$^{-1}$ and in conventional with intensive protection 49.1 t·ha$^{-1}$ Significantly higher tuber yields were achieved in 2011. Differences in tuber yield in individual production systems, however, depended on the years of research. In the organic system, despite different pest pressure, the yield was at a similar level in both years. In the conventional system, and especially those with intensive protection, in 2011, tuber yield was significantly higher than in 2010. It should be noted that in 2010, with low pathogen pressure, the difference in tuber yield between the conventional system with limited protection was the 17 t·ha$^{-1}$, and in the "blight" year, i.e., 2011, as much as 24.5 t·ha$^{-1}$.

Comparing the rate of spread of late blight in individual production systems and tuber yield, a strict inverse relationship was found, i.e., the faster the development of the disease in a given system, the lower the yield. On average, for two years of research, the highest rate of spreading the disease and, at the same time, the lowest yield of tubers were recorded in the organic system. In the conventional system with intensive protection, the rate of pathogen spread was the slowest and the yield of tubers was the highest. The increase in yield under the influence of treatments limiting the development of late blight in our studies ranged from 10 to 15%.

As can be seen from the presented results, the resistance of cultivars to *P. infestans* is of key importance. Kapsa research [15] shows that cultivars having a resistance of 7 to 8 on a 9-point scale can protect themselves from infection, especially in the late onset of the disease. Our research conducted in the years 2008–2010 confirmed the relationship between the resistance of the cultivar and the development of late blight. The Ursus variety with resistance 6.5 had almost six times slower rate of disease development than the Miłek variety with resistance 3 (Figure 8).

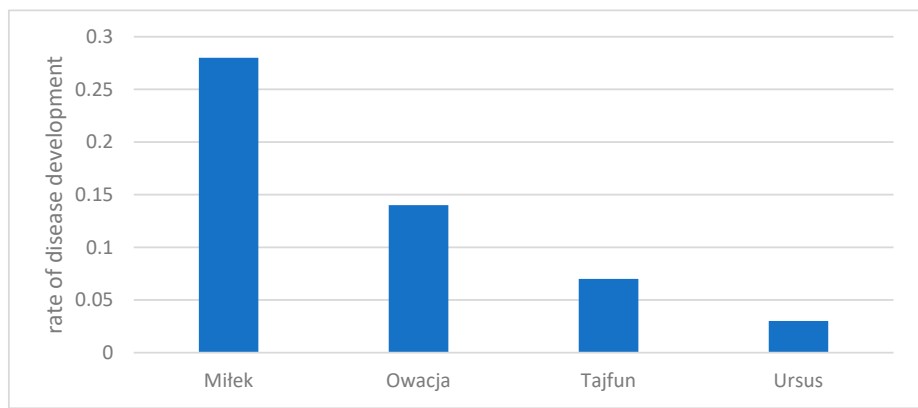

**Figure 8.** Rate of late blight development depending on cultivars resistance to *P. infestans* (Jadwisin 2008–2010).

Many authors confirm similar relationships [15,87–91].

The demand for cultivars with increased resistance to *P. infestans* is increasing. Potato breeding companies are already successful in this field. In recent years, a highly resistant cultivar Gardena has been bred in Poland. The resistance of this cultivar to *P. infestans* is 7 out of 9. This cultivar was tested in our research in comparison to a cultivar commonly cultivated in Poland in organic farming—Denar with resistance 3. The spread rate of late blight in mid-August 2021 was 0.119 for the Denar cultivar and the disease did not progress in the Gardena cultivar (authors' observation).

In addition to the above-mentioned agrotechnical treatments limiting the development of late blight, alternative methods of controlling late blight are still being sought. Research on biological methods using the antagonistic effects of bacteria and fungi in combating late blight is being conducted all the time [92]. Positive results are most often obtained in laboratory tests. In field conditions, these effects are generally small. Two factors, probably among others, that make biocontrol difficult to this disease are rapid establishment of infection and explosive disease development. It is reasonable to assume that many attempts to use biocontrol for potato late blight have been unsuccessful and this may be the reason why the literature in this field is so scarce [93].

In our research, Trichoderma fungus was used, which slightly limited the spread of the disease. In the research of Kowalska et al. [94], in field trials, a possibility of limitation of potato late blight by *Trichoderma asperellum* as well as its influence on vitality of plants and yielding was assessed. In experiments the tested fungus to soil and leaves was applied. Degree of infection was noted as area of infected parts of plant. One application to the soil and four foliar treatments resulted in the efficacy comparable with two copper treatments. Many foliar treatments (10×) reduced the infected area of the plant by 30% compared to the level of infection on untreated plants. The yield from microbial treated fields revealed a higher number of small tubers, the total yield was significantly higher.

Of the remaining agronomic factors combating late blight, the destruction of sources of infection and the use of healthy seed potatoes are also emphasized [95,96].

## 4. Main Conclusions

Summarizing our own and other research on the topic in question, the following conclusions can be drawn:

The yield of potatoes grown in the organic system depends greatly on weather conditions in the vegetation period and cultivar. The highest yields are obtained in the years of the best distribution of rainfall during the growing season (provided effective protection against late blight). In the years of unfavorable weather conditions, more significant losses in yield can be expected in the organic than conventional farms.

There are very large cultivar differences in tuber yield. Most cultivars yielded in the range of 20–30 t·ha$^{-1}$. The proper selection of cultivar for the conditions prevailing in a given area should be taken into account. The domestic varieties seem to be better for cultivation in the organic system.

A weaker development of potato plants in the organic system results in a lower yield and a smaller share of marketable tubers. The advantage of slower plant development is a later date of appearance of late blight and a slower rate of its development. It is possible to increase the yield of potato tubers grown in the organic system through cultivation treatments such as accelerating vegetation through presprouting of seed potatoes, the use of complementary fertilization allowed in organic farming, irrigation of the plantation with the use of drip irrigation, and the use of agronomical treatments limiting the development of late blight. In our research, drip irrigation increased the yield by 25.5%, complementary fertilization by up to 19%, agrotechnical treatments limiting the development of late blight by 15%, and presprouting of seed potatoes by 4.3%. Most agronomical treatments not only improved the total yield of tubers, but also increased the share of tubers with a larger diameter.

**Author Contributions:** Conceptualization, K.Z.; methodology, K.Z. and C.T.; investigation, K.Z.; data curation, K.Z. and C.T.; writing—original draft preparation, K.Z.; writing—review and editing, K.Z.; visualization, M.P.; supervision, K.Z.; project administration, K.Z.; funding acquisition, K.Z. and C.T. All authors have read and agreed to the published version of the manuscript.

**Funding:** Polish Ministry of Agriculture and Rural Development, DC, IHAR-PIB (6):3.

**Institutional Review Board Statement:** Not applicable.

**Informed Consent Statement:** Not applicable.

**Data Availability Statement:** The data presented in this study are available upon request from the first author.

**Conflicts of Interest:** The authors declare no conflict of interest. The funders has no role in the design of the study; in the collection, analyses or interpretation of data; in the writing of the manuscript; or in the decision to publish the results.

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
