# Peer review of "Environmental and Agronomical Factors Limiting Differences in Potato Yielding between Organic and Conventional Production System"

_agriculture, doi:10.3390/agriculture13040901_

Round 1

Reviewer 1 Report

Dear Authors

Your field of study and your aim is good. But you have some big problem with writing and structure in all sections of the manuscript. I have mentioned some important note in the file and would like to mention some note here. 

1- Please, rewrite the introduction 

2- Your M&M is not clear. Please rewrite and attention to my comment at end of M&M.

3-potato is a crop expensive management, production system. If you can evaluate the economic factors, to your work would be better than now. 

Other comments are in the attachment file.

Best regards,

Author Response

The answer is attached below

Reviewer 2 Report

Dear authors,

Thank you for the opportunity to review this Manuscript (Environmental and agronomical factors limiting differences in potato yielding between organic and conventional production system). The study has great results and demonstrates the results of own and literature research on the impact of selected environmental and agronomical factors on the yield of potato grown under the organic system and the possibility of increasing the yield. The aim of the work was to indicate the environmental and agrotechnical factors affecting the yield of potato tubers grown in the organic system and to show which of these factors can reduce the differences in yield between the organic and conventional systems. There is some aspect that should be reviewed by authors, but the Manuscript is well-written.

ABSTRACCT

Give more information of the field experiment

Give data information

Add data as example in the results.

INTRODUCTION

The authors could explore and give more details about organic system

Also, more information about the conventional systems.

MATERIAL AND METHODS + RESULTS AND DISCUSSION

This topic is very superficial.

Add information about the soil, climate and management of potato.

Genetic material of potato

All Figures have low quality

Add titles in Figures

The authors made some confusion to explain how is “Material and Methods” e “Results and discussion”

Add experimental design

Figure 5, Is it treatments?

The authors should made clear is it results from study or from the literature.

The conclusion should compare the yield systems.

Round 2

Reviewer 1 Report

Dear Authors

Please send PCA picture. i would like to review that. the same as previous comments i believe that your M&M need more details about location, data analysis approaches ,...

Also, your results and discussion must be separated

Author Response

Response below

Reviewer 2 Report

Poor-quality images, please check it.

Author Response

Response below
